# Exploring the potential effect of paricalcitol on markers of inflammation in *de novo* renal transplant recipients

Hege Kampen Pihlstrøm[1]*, Thor Ueland[2,3,4], Annika E. Michelsen[2,3], Pål Aukrust[2,3,5], Franscesca Gatti[6,7], Clara Hammarström[6,7], Monika Kasprzycka[6,7], Junbai Wang[6], Guttorm Haraldsen[6,7], Geir Mjøen[1], Dag Olav Dahle[1], Karsten Midtvedt[1], Ivar Anders Eide[8], Anders Hartmann[1,3], Hallvard Holdaas[1]

1 Department of Surgery, Inflammation Medicine and Transplantation, Section of Nephrology, Oslo University Hospital, Rikshospitalet, Oslo, Norway, 2 Research Institute of Internal Medicine, Oslo University Hospital, Rikshospitalet, Oslo, Norway, 3 Institute of Clinical Medicine, Faculty of Medicine, University of Oslo, Oslo, Norway, 4 K.G. Jebsen Thrombosis Research and Expertise Center, University of Tromsø, Tromsø, Norway, 5 Department of Surgery, Inflammation Medicine and Transplantation, Section of Clinical Immunology and Infectious Diseases, Oslo University Hospital, Rikshospitalet, Oslo, Norway, 6 Department of Pathology, Oslo University Hospital, Oslo, Norway, 7 K.G. Jebsen Inflammation Research Centre, Laboratory of Immunohistochemistry and Immunopathology, University of Oslo, Oslo, Norway, 8 Division of Medicine, Department of Nephrology, Akershus University Hospital, Oslo, Norway

* hegphi@ous-hf.no

**Data Availability Statement:** Norwegian Data Protection Authorities have a strict practice concerning access to research data. Access to data collected from this study, including de-identified

## Abstract

Following a successful renal transplantation circulating markers of inflammation may remain elevated, and systemic inflammation is associated with worse clinical outcome in renal transplant recipients (RTRs). Vitamin D-receptor (VDR) activation is postulated to modulate inflammation and endothelial function. We aimed to explore if a synthetic vitamin D, paricalcitol, could influence systemic inflammation and immune activation in RTRs. Newly transplanted RTRs were included in an open-label randomized controlled trial on the effect of paricalcitol on top of standard care over the first post-transplant year. Fourteen pre-defined circulating biomarkers reflecting leukocyte activation, endothelial activation, fibrosis and general inflammatory burden were analyzed in 74 RTRs at 8 weeks (baseline) and 1 year post-engraftment. Mean changes in plasma biomarker concentrations were compared by *t*-test. The expression of genes coding for the same biomarkers were investigated in 1-year surveillance graft biopsies (n = 60). In patients treated with paricalcitol circulating osteoprotegerin levels increased by 0.19 ng/ml, compared with a 0.05 ng/ml increase in controls (p = 0.030). In graft tissue, a 21% higher median gene expression level of TNFRSF11B coding for osteoprotegerin was found in paricalcitol-treated patients compared with controls (p = 0.026). Paricalcitol treatment did not significantly affect the blood- or tissue levels of any other investigated inflammatory marker. In RTRs, paricalcitol treatment might increase both circulating and tissue levels of osteoprotegerin, a modulator of calcification, but potential anti-inflammatory treatment effects in RTRs are likely very modest.

[NCT01694160 (2012/107D)]; [www.clinicaltrials.gov].

individual participant data, will be made available following publication upon email request to the Institutiona Data Protection Officer. Data will be shared with researchers who meet the criteria for access to confidential data. The data underlying the results presented in the study are available from the OUS Data Protection Officer, Tor Åsmund Martinsen (personvern@oslo-universitetssykehus. no). The trial protocol is available as a supplement to this manuscript. Fully anonymized microarray data have been made accessible through the GEO database (Accession No. GSE83486; http://www. ncbi.nlm.nih.gov/geo/query/acc.cgi?acc= GSE83486).

**Funding:** The trial was funded via a PhD grant to Hege K Pihlstrøm from South Eastern Norway Regional Health Authority (https://www.helse-sorost.no) and received an additional research grant from the Norwegian Society of Nephrology (https://nephro.no). Norwegian Health Authorities (https://helfo.no) also funded the trial by covering all expenses associated with paricalcitol treatment. The funders had no role in study design, data collection and analysis, decision to publish, or preparation of the manuscript.

**Competing interests:** The authors have declared that no competing interests exist.

**Abbreviations:** CI, confidence interval; CKD, chronic kidney disease; CNI, calcineurin inhibitor; CRP, C-reactive protein; DLL1, delta like canonical Notch ligand 1; ECM, extracellular matrix; ELISA, enzyme-linked immunosorbent assay; eGFR, estimated glomerular filtration rate; hsCRP, high-sensitive C-reactive protein; IL-6, interleukin-6; MMP-9, matrix metalloprotease-9; mRNA, messenger ribonucleic acid; NGAL, neutrophil gelatinase-associated lipocalin; OPG, osteoprotegerin; PAT, peripheral arterial tonometry; PTH, parathyroid hormone; RAAS, renin-angiotensin-aldosterone system; RANK, Receptor activator of nuclear factor kappa-B; RANKL, Receptor activator of nuclear factor kappa-B ligand; RTRs, renal transplant recipients; SD, standard deviation; TGF-β, transforming growth factor beta; TIMP-1, tissue inhibitor of metalloproteinase -1; TNF, tumor necrosis factor; vWF, von Willebrand factor; VDRA, vitamin D receptor agonist.

# Introduction

Chronic kidney disease (CKD) is associated with systemic inflammation. In end-stage renal disease there is a strong association between inflammation marker C-reactive protein (CRP) and risk of death and cardiovascular events [1]. Increased oxidative stress, the accumulation of toxic metabolites (e.g. microbiota-dependent amine oxides) and chronic activation of various cell subsets of both the innate and adaptive immune system creates a pro-inflammatory milieu in CKD [2, 3]. Following a successful renal transplantation with appropriate immunosuppression obtained by a combination of drugs, both endothelial dysfunction and low-grade inflammation is ameliorated by reversal of the uremic state [4, 5], while the lipid oxidative state seems more refractory [6]. Some derangements in immune function will however persist [7, 8], especially if graft function is sub-optimal. Even among renal transplant recipients (RTRs) with low Framingham risk scores, implying a limited burden of comorbidity, there is evidence for enhanced systemic inflammation [9]. In stable RTRs, interleukin-6 (IL-6) and CRP, has been associated with risk of major cardiovascular events and all-cause mortality [10], as has neopterin [11, 12], a marker of interferon (IFN) γ-mediated activation of monocytes/macrophages.

Vitamin D is a fat-soluble vitamin central to the maintenance of bone- and mineral homeostasis. Poor vitamin D status has also been associated with increased risk of cancer [13], infections [14, 15], autoimmune diseases [16], cardiovascular events [17], obesity [18] and diabetes [19]. However, interventional studies on vitamin D supplementation report equivocal and inconsistent effects on non-skeletal clinical outcomes, and larger randomized controlled trials are ongoing [20–22]. Anti-inflammatory and immune-modulating effects of vitamin D -supplements could be of particular benefit for renal transplant patients, as they are prone to vitamin D-deficiency, while at the same time carrying an increased risk of all of the above mentioned chronic conditions [23, 24].

Paricalcitol (19-nor-1,25-dihydroxyvitamin D2), a synthetic selective third generation vitamin D-receptor agonist (VDRA), is used in patients with CKD to treat secondary hyperparathyroidism. Potential non-skeletal benefits of paricalcitol include anti-proteinuric effects in patients with diabetes [25] and in RTRs [26]. VDRAs seem also to have anti-inflammatory potential, and experimental studies have indicated dampened tumor necrosis factor (TNF) and interleukin-8 (IL-8) production [27] and reduced inflammation and fibrosis development [28].

An observational study in RTRs with secondary hyperparathyroidism found that 3 months of treatment with paricalcitol reduced serum IL-6 and TNF levels, with corresponding lowered mRNA expression in peripheral blood mononuclear cells [29]. In a clinical trial of 168 RTRs with proteinuria, paricalcitol treatment on top of RAAS-blockade caused significant reductions in circulating IL-6 and the pro-fibrotic mediator transforming growth factor beta (TGF-β) [30]. We were, however, not able to demonstrate a treatment effect of paricalcitol on high sensitivity (hs) CRP in 77 newly transplanted RTRs [31], but it is unlikely that hsCRP reflects all inflammatory pathways that are activated in RTRs.

Cytokines and inflammatory molecules are operating in a complex network, and our aim in the current study on renal transplant recipients was to explore the potential effect of paricalcitol on a broader range of pre-defined circulating inflammatory markers, including markers reflecting leukocyte activation, endothelial activation, fibrosis and more general vascular inflammation.

# Materials and methods

## Study design

From Jan 2013 up until Jan 2014, 77 patients >18 years of age who had received a kidney transplant or a combined kidney-pancreas transplant were randomized to receive either

treatment with paricalcitol 2 μg daily or standard post-transplant care. Study randomization and baseline laboratory measurements took place 7–8 weeks after transplantation. The last study visit was scheduled one year after date of engraftment and included repeated blood sampling and collection of renal transplant biopsy tissue for subsequent histopathology evaluation and RNA extraction. To be eligible for the study, patients should have an estimated glomerular filtration rate (eGFR) by the CKD-EPI formula of at least 30 ml/min, total serum calcium levels should range between 2.0 and 2.6 mmol/l, and the immunosuppressive regimen should include a calcineurin inhibitor (CNI). Patients already undergoing treatment with vitamin D, VDRA or calcimimetic drugs were excluded, as were patients with established osteoporosis in the axial skeleton, patient with a history of hypersensitivity towards paricalcitol or related drugs and recipients receiving organs from a donor older than 75 years. Results concerning the primary endpoint (potential anti-proteinuric effect) and main secondary endpoints are found elsewhere [31], as is the Consort diagram presenting the screening- and inclusion process. In short, both the baseline- and 1-year visits consisted of the following investigations: Blood samples for routine biochemistry and frozen storage, spot urine albumin/creatinine ratio, pulse wave velocity measurements, assessment of endothelial function by a noninvasive plethysmographic method, allograft protocol biopsy with collection of tissue for RNA extraction and measured glomerular filtration rate (mGFR) by iohexol clearance. All patients provided written informed consent before inclusion in the trial. The study conformed to the principles of the Declaration of Helsinki and the Declaration of Istanbul. The study protocol was approved by the Regional Ethics Committee, officially known as REK South East (study no 2012/107) and the hospital´s Research Administration (The Oslo University Hospital Data Protection Authority as well as the Radiology Research Administration, FU-ARN). It was also approved by the Norwegian Medicines Agency, SLV (Eudract no: 2012-000429-32). The Department of Organ Transplantation at Oslo University Hospital was responsible for the coordination and conduction of the trial.

## Outcomes

Cytokines and inflammatory molecules are operating in a complex network, and our aim in the current study was to explore the potential effect of paricalcitol on a broader range of stable and readily measurable circulating inflammatory markers. We included biomarkers reflecting *leukocyte activation* (neopterin, soluble CD14 and soluble CD163 as markers of monocyte/macrophage activation; neutrophil gelatinase-associated lipocalin [NGAL] as a marker of neutrophil activation), *endothelial activation* (von Willebrand factor [vWf], and angiopoietin-2), *fibrosis* (endostatin, matrix metalloprotease-9 [MMP-9], galectin3, tissue inhibitor of metalloproteinase 1 [TIMP-1], activin A) and more *general vascular inflammation* (osteoprotegerin [OPG]), soluble tumor necrosis factor-receptor 1 [TNFR 1] and delta like canonical Notch ligand 1 [DLL1]). As a supplementary analysis the levels of mRNA reflecting expression of genes coding for the same biomarkers were investigated in renal graft tissue at study end.

## Laboratory

After an overnight fast, blood samples were drawn in the morning at the baseline visit and at the last study visit one year after transplantation. After centrifugation at 2350 g for 10 minutes, plasma and serum were immediately frozen and samples stored at -72˚ C until analysis (April-May 2018). Plasma levels of inflammatory markers were measured in duplicate by enzyme immunoassay (EIA) using R&D Systems (Stillwater, Minneapolis, MN) antibody pairs: Angio-poietin-2 (DY623), sCD14 (DY383), sCD163 (DY1607), DLL1 (DY1818), Endostatin (DY1098), MMP9 (DY911), sTNFr1(DY225), Galectin-3 (DY1154), NGAL (DY1757),

ActivinA (DY338), OPG (DY805), TIMP-1 (DY970). For von Willebrand factor (VWF) the EIA was performed with antibodies (A0082, P0226) from DakoCytomation (Glostrup, Denmark) and for Neopterin, a kit (RE59321) from IBL International GmbH (Hamburg, Germany). Assays were performed in a 384-format using the combination of a SELMA (Jena, Germany) pipetting robot and a BioTek (Winooski, VT) dispenser/washer (EL406). Primary and secondary antibody concentrations were used according to manufacturer instructions (Coating 1–4 μg/ml; secondary 0.2–2 μg/ml). Assay volume was 20 μl and coating was performed in phosphate buffered saline. Subsequent assay buffer was with 1% bovine serum albumin in PBS while sample diluent was PBS with 25% heat inactivated fetal calf serum (Gibco, Thermo Fisher Scientific, Waltham, MA). Wash buffer was PBS with 0.05% tween-20 and three wash cycles were included per step. Samples were incubated overnight at 4˚C. Absorption was read at 450 nm with wavelength correction set to 540 nm using an EIA plate reader (Synergy H1 Hybrid, Biotek, Winooski, VT). Intra- and interassay coefficients of variation were <10% for all assays. The assays included a series of known concentrations to generate standard curves.

## Gene expression analyses

The procedure of RNA extraction from retrieved transplant biopsy tissue stored in RNA*later*® solution, together with RNA quality assessment, amplification- and labelling procedures are described elsewhere [31]. Sixty samples, equally divided between study groups, were found to have sufficient quality for microarray gene expression analyses. For this post-hoc investigation, we selected 15 gene products reflecting the expression of 13 biomarker proteins. (Neopterin is a degradation product for which the circulating level is unlikely to be directly reflected by the expression any gene).

## Statistical methods

The intention-to-treat population consisted of any patient who was randomized and, if assigned to the treatment group, received at least one dose of study drug, irrespective of any study protocol violation. The per-protocol population consisted of participants actually fulfilling the protocol requirements for eligibility, intervention and outcome assessment.

Comparisons of baseline variables between study groups were done using *t*-test, Mann-Whitney U Test or Pearson χ2 as found appropriate. The potential correlation between change in osteoprotegerin and change in PTH was tested using Pearsons Correlation test.

The levels of biomarkers in the circulation, expressed as *change from baseline to study end*, showed only marginal deviations from a normal distribution, rendering the *t*-test for independent observations applicable. As a sensitivity analysis, ANCOVA was performed, analyzing group differences in levels of biomarkers at study end with adjustments for baseline levels.

For analyses of microarray data, gene expression levels were log transformed, and normalized intensities were converted to Z-scores, which were used to identify differentially expressed genes between the paricalcitol group and controls. For each gene, a relative ratio of the mean Z-scores between the two groups was computed, and the statistical significance of relative ratios (P-value) was estimated by the two-sample Kolmogorov-Smirnov goodness-of-fit hypothesis test (KS-test), which does not have a prior assumption for the distribution of gene expression [32].

Statistical analyses of circulating biomarker levels were performed in SPSS version 21 (IBM, New York, USA). Microarray gene expression data analysis were performed by either MATLAB statistics toolbox (MathWorks, Natick, USA) or in-house script files such as Python based on previously published works [33].

# Results

## Baseline characteristics

A total of 74 patients had available plasma samples for analyses of circulating biomarkers at baseline and study end. Baseline demographics, as well as relevant laboratory values and vital signs are presented for both study groups in Table 1. There were no statistically significant differences between groups, but there was a trend towards more males (Pearson χ2, p = 0.19) and

**Table 1. Baseline characteristics of the study population[a].**

| Variables | Paricalcitol | Control |
|---|---|---|
|  | (n = 35) | (n = 39) |
| Age, years | 55.6 (13.3) | 55.1 (12.6) |
| Male gender | 26 (74%) | 33 (85%) |
| Caucasian ethnicity | 34 (97%) | 37 (95%) |
| BMI, kg/m$^2$ | 26.2 (3.3) | 25.5 (3.9) |
| Current smoking | 5 (14%) | 5 (13%) |
| Living donor | 10 (29%) | 12 (31%) |
| Cold ischemia time, hours | 10.4 (6.4) | 10.1 (5.7) |
| Glomerulonephritis as cause of CKD | 13 (37.1) | 15 (38.5) |
| Predialytic | 13 (37%) | 13 (33%) |
| Hypertension | 29 (83%) | 36 (92%) |
| Chronic heart disease | 11 (31%) | 13 (33%) |
| Pre-tx diabetes mellitus | 6 (17%) | 6 (15%) |
| Systolic blood pressure, mmHg | 145 (21) | 143 (22) |
| Diastolic blood pressure, mmHg | 83 (10) | 84 (11) |
| Treatment with ACEi/ARB, % | 9 (26%) | 14 (36%) |
| Cholesterol, mmol/L | 5.8 (1.1) | 5.9 (0.9) |
| HDL cholesterol, mmol/L | 1.6 (0.5) | 1.6 (0.4) |
| LDL cholesterol, mmol/L | 3.8 (1.0) | 3.9 (0.9) |
| Triglycerides, mmol/L* | 1.3 (1.0) | 1.4 (0.5) |
| Creatinine, μmol/L | 115 (25) | 122 (30) |
| Hemoglobin, g/L | 12.4 (1.2) | 12.3 (1.2) |
| hsCRP, mg/L ¤* | 0.85 (2.20) | 1.00 (1.19) |
| Calcium total, mmol/L | 2.38 (0.09) | 2.34 (0.21) |
| Phosphate, mmol/L * | 0.9 (0.3) | 0.8 (0.4) |
| Albumin, g/L | 42.3 (2.5) | 41.5 (2.4) |
| PTH, pmol/L * | 10.1 (9.2) | 10.2 (5.4) |
| Alkaline phosphatase, U/L | 60.7 (21.8) | 69.4 (28.6) |
| Vitamin 25-OH-D, nmol/l | 50.1 (18.0) | 44.8 (17.2) |
| Urine Albumin/creatinine ratio, mg/mmol * | 3.1 (7.4) | 4.5 (8.7) |

BMI, body mass index; CKD, chronic kidney disease; HDL, high density lipoprotein; hsCRP, high-sensitive C-reactive protein; LDL, low density lipoprotein; PTH, parathyroid hormone; ACEi, angiotensin converting enzyme inhibitor; ARB, angiotensin receptor blocker.

All laboratory measurements are performed in plasma.

[a]Modified version of table from the original publication [31]. Continuous data expressed as mean (standard deviation) or * median (interquartile range).

Categorical data expressed as number (percentage frequency).

¤ Values <0.60mg/L (laboratory detection cut-off) are all given the value 0.30. Values >15mg/L are rounded down to this value.

a lower baseline vitamin 25-OH-D (*t*-test, p = 0.16) in the control group. Twenty-eight patients (38%), equally divided between study groups, had an immunological cause of end-stage renal disease, e.g. glomerulonephritis.

Two patients were diagnosed with biopsy-proven antibody mediated rejection at 1-year follow up, while four patients were being treated for acute cellular rejection at time of study inclusion or earlier postoperatively. Two patients suffered a cellular rejection in the interval between study visits. Seven patients suffered one or more infections needing systemic antibiotic treatment over the duration of the study. None of these participants have been excluded from the primary efficacy analyses.

## Effects of paricalcitol on serum levels of inflammatory markers

Paricalcitol did not significantly reduce the levels of measured inflammatory markers as compared with no treatment. Table 2 presents intention-to-treat analyses, showing the mean (or

**Table 2. Plasma levels of biomarkers at baseline and study end, by treatment group.**

| Biomarker(plasma levels) | Paricalcitol (n = 35) | | | Control (n = 39) | | | *t*-test | ANCOVA |
|---|---|---|---|---|---|---|---|---|
| | Baseline Mean (SD) | 1-year Mean (SD) | Change (%) | Baseline Mean (SD) | 1 year Mean (SD) | Change (%) | p-values (CI ng/ml) for group differences in change | p-values (CI ng/ml) |
| Angiopoietin-2 (ng/ml)* | 0.74 (0.49) | 0.69 (0.41) | -6.8 | 0.67 (0.46) | 0.72 (0.55) | +7.5 | 0.478 (-0.57–0.27) | 0.561 (-0.52–0.29) |
| sCD14 (ng/ml) | 1.57 (0.21) | 1.57 (0.30) | +0.0 | 1.58 (0.27) | 1.52 (0.29) | -3.8 | 0.383 (-0.08–0.21) | 0.375 (-0.07–0.19) |
| sCD163 (ng/ml) | 441 (230) | 547 (316) | +24.0 | 467 (238) | 512 (190) | +9.6 | 0.241 (-42.3–166) | 0.287 (-45.0–150) |
| DLL1 (ng/ml) | 8.88 (2.07) | 9.73 (2.83) | +9.6 | 9.62 (2.40) | 9.85 (2.84) | +2.4 | 0.212 (-0.37–1.62) | 0.312 (-0.49–1.51) |
| Endostatin (ng/ml) | 99.1 (21.8) | 93.4 (21.6) | -5.8 | 108.7 (32.2) | 103.0 (32.1) | -5.2 | 0.908 (-9.9–11.2) | 0.593 (-12.5–7.2) |
| MMP9 (ng/ml) | 99.8 (67.7) | 74.4 (43.7) | -25.4 | 77.4 (35.9) | 79.7 (78.8) | +2.3 | 0.058 (-56.3–0.95) | 0.357 (-31.6–11.5) |
| sTNFr1(ng/ml) | 1.95 (0.67) | 2.01 (0.91) | +3.1 | 2.22 (0.78) | 2.16 (0.79) | -2.7 | 0.464 (-0.21–0.46) | 0.809 (-0.29–0.37) |
| Galectin-3 (ng/ml) | 1.41 (0.56) | 1.17 (0.55) | -17.1 | 1.48 (0.57) | 1.26 (0.44) | -14.9 | 0.835 (-0.25–0.20) | 0.543 (-0.24–0.13) |
| NGAL (ng/ml) | 282 (122) | 312 (144) | +10.6 | 287 (145) | 305 (133) | +6.5 | 0.735 (-52.0–73.3) | 0.681 (-48.6–73.9) |
| vWF in % of ref. plasma* | 82.5 (89.5) | 57.1(37.0) | -30.8 | 102.8 (118.3) | 57.8 (58.6) | -43.8 | 0.228 (-15.1–62.3) | 0.215 (-8.6–37.4) |
| ActivinA (ng/ml)* | 344 (739) | 323 (1075) | -6.1 | 578 (910) | 681 (873) | +17.8 | 0.696 (-206–306) | 0.778 (-227–302) |
| OPG (ng/ml) | 0.91 (0.37) | 1.10 (0.44) | +20.9 | 1.08 (0.53) | 1.13 (0.52) | +4.6 | 0.030 (0.01–0.26) | 0.062 (-0.01–0.24) |
| TIMP-1 (ng/ml) | 110 (23) | 112 (26) | +1.8 | 124 (32) | 122 (38) | -1.6 | 0.461 (-7.2–15.6) | 0.808 (-10.2–13.0) |
| Neopterin (nmol/L)* | 21.4 (18.5) | 21.1 (13.8) | +1.4 | 22.1 (13.7) | 20.4 (11.4) | +7.7 | 0.787 (-12.5–16.3) | 0.169 (-2.2–12.04) |

Intention-to-treat population. *T*-test for difference in change and supplementary ANCOVA: p-values presented with corresponding confidence intervals (CI).

DLL1, delta like canonical Notch ligand 1; MMP9, matrix metalloprotease-9; sTNFR1, soluble tumor necrosis factor receptor-1; NGAL, neutrophil gelatinase-associated lipocalin; vWF, von Willebrand factor; OPG, osteoprotegerin; TIMP-1, Tissue inhibitor of metalloproteinase 1.

Data expressed as mean (standard deviation) or * median (interquartile range).

Continuous data expressed as mean (standard deviation) or * median (interquartile range).

median) levels of inflammatory biomarkers in each group at 8 weeks and 1 year post transplant, together with the changes from baseline to study end given in percent of baseline mean (or median). *T*-test for likelihood of observing the reported results given no true group difference is presented with corresponding p-values and confidence interval for the absolute group difference in change. MMP-9 levels decreased in the paricalcitol group, while it increased in the control group, but the difference was not statistically significant (*t*-test for difference in change [ng/ml], p = 0.058; CI -56.3–0.95). Paricalcitol treatment was, however, associated with *increased* mean OPG levels, as opposed to nearly no change in the control group (*t*-test for difference in change [ng/ml], p = 0.030; CI 0.01–0.26). A few extreme values were responsible for at least some of this difference (Fig 1). For all other parameters, p-values for differences in changes were >0.1 and potential effect sizes were small (Table 2).

No significant correlation was found between change in osteoprotegerin and change in PTH (Pearson's Correlation test, p = 0.74). Sensitivity analyses using ANCOVA did not materially change results: p = 0.062 for difference between groups in OPG at 1-year adjusted for baseline OPG level (rightmost column Table 2).

Analyses of the per-protocol population (n = 67) did not change results; p = 0.041 for increase in OPG and p = 0.076 for reductions in MMP-9 with paricalcitol treatment (*t*-test, S1 Table). Results of sensitivity analyses excluding patients diagnosed with rejection at any time

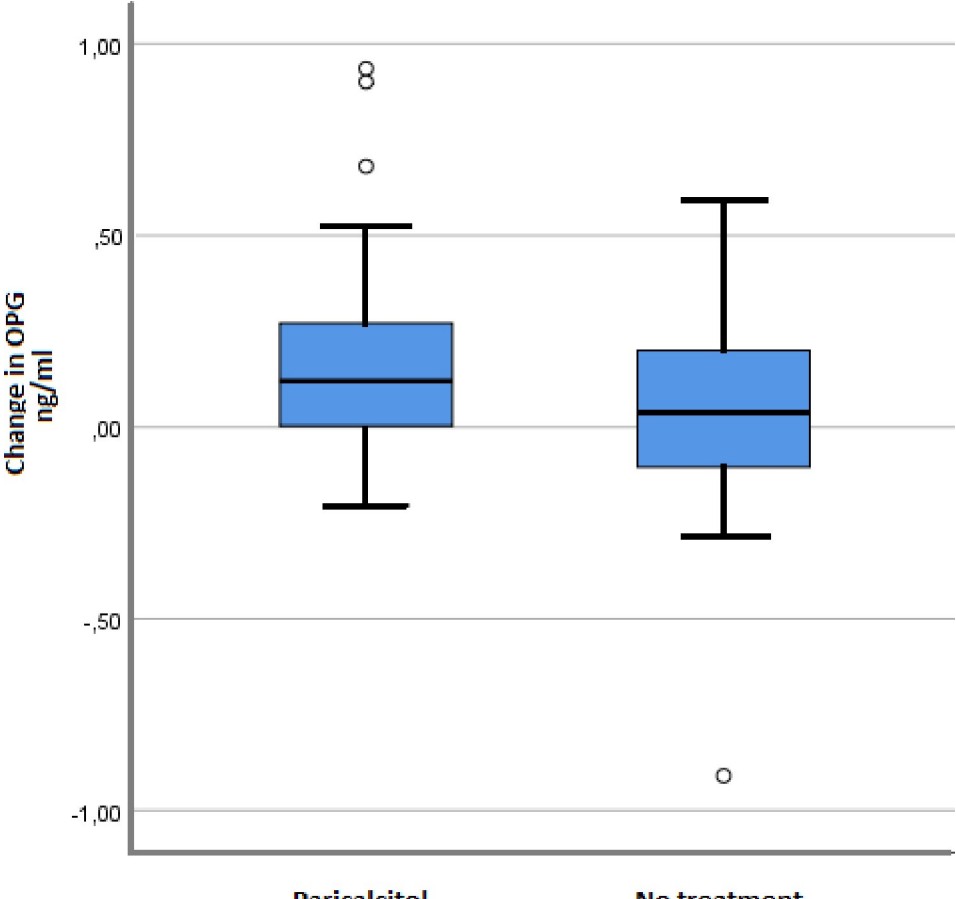

**Fig 1. Changes in levels of osteoprotegerin across the study period.** Osteoprotegerin change (ng/nl) in patients treated with paricalcitol vs patients receiving no extra treatment; median (horizontal line), interquartile range (blue box), outlier (˚).

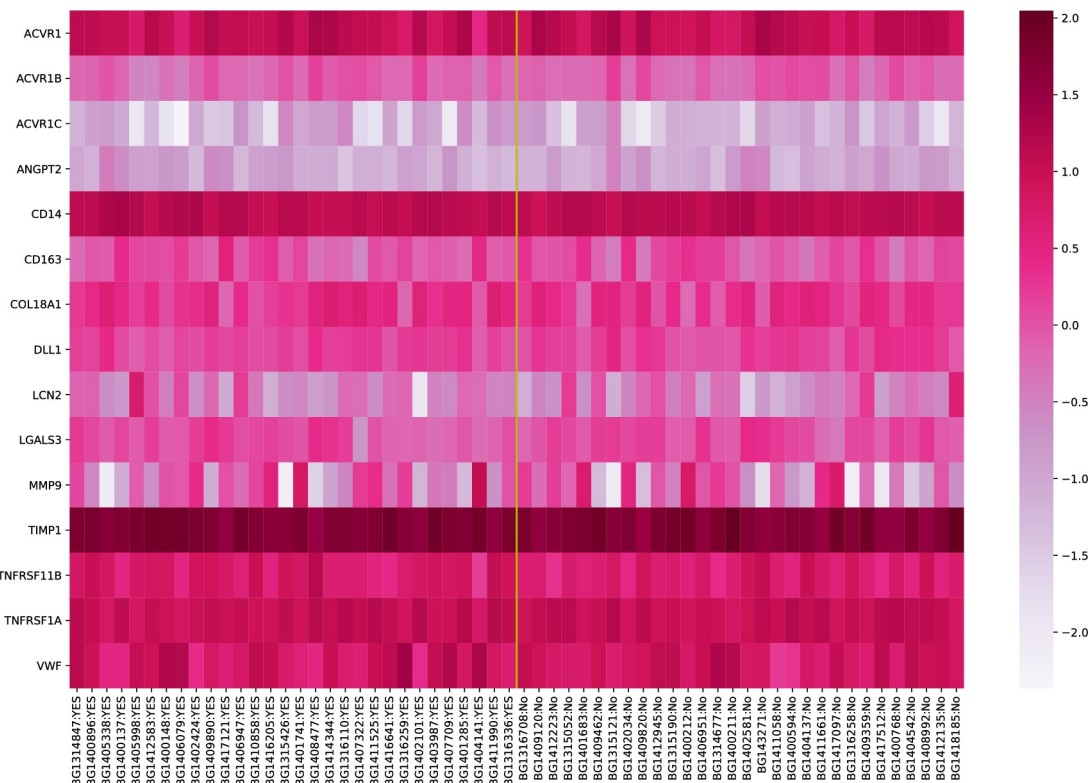

**Fig 2. Heat map of inflammatory marker gene expression levels in graft tissue.** The expression of 15 genes coding for 13 inflammatory biomarkers in 30 treated patients (to the left) vs 30 patients in the control group (to the right). Darker color indicates higher expression levels. Z-scores of duplicated genes in the array are averaged. Genes coding for proteins with different nomenclature: ACVR1/ACVR1B/ ACVR1C, activin A receptor subunits; ANGPT2, angiopoietin-2; COL18A1, endostatin; LCN2, neutrophil gelatinase-associated lipocalin (NGAL); LGALS3, galectin-3; TNFRSF11B, osteoprotegerin; TNFRSF1A, soluble tumor necrosis factor receptor-1 (sTNFr1).

point during follow up (n = 67) were also comparable to the main analysis; p = 0.048 for increase in OPG and p = 0.086 for reductions in MMP-9 (*t*-test, S2 Table). Exclusion of patients with glomerulonephritis as cause of ESRD did not affect results; n = 46, OPG: p = 0.054 and MMP-9: p = 0.018 (*t*-test, S2 Table).

## Gene expression in renal graft tissue in response to paricalcitol

In renal graft tissue of patients treated with paricalcitol, there was a 21% higher expression of TNFRSF11B, the gene coding for osteoprotegerin, compared with the control group (median gene expression 0.808 vs 0.668; p = 0.026 by KS test). We detected no other significant differences in the expression of biomarker genes between patients treated with paricalcitol and controls, as illustrated by the heat map (Fig 2). All microarray data are available at the Gene Expression Omnibus (GEO) database; http://www.ncbi.nlm.nih.gov/geo/query/acc.cgi?acc=GSE83486.

## Discussion

We explored the potential effects of paricalcitol treatment during the first year after kidney transplantation on a wide range of biomarkers reflecting several aspects of inflammatory responses, but were unable to confirm clinically or statistically significant effects ameliorating inflammation. Paricalcitol treatment did, however, increase circulating OPG levels.

Importantly, this was accompanied by a corresponding increase in TNFRSF11b, the gene coding for OPG, in biopsies from renal allografts, supporting a link between paricalcitol and OPG.

The proposed anti-inflammatory effects of paricalcitol are ascribed to a regulatory role of vitamin D receptor activation in several subsets of immune cells, such as macrophages, dendritic cells and T-cells [34, 35]. However, in one clinical study in patients with vascular inflammation, paricalcitol appeared to exert its effect rather selectively on T-helper cells by interfering with calcineurin-mediated responses [36]. Modulation of adaptive immune responses in CKD-patients has been demonstrated, reflected by reductions in several inflammatory markers of Th1- Th2 and Th17-responses [37]. Oblak et al. [30] demonstrated treatment effects in a quite large interventional study of RTRs. However, most clinical trials suggesting anti-inflammatory properties of paricalcitol in CKD-patients [38–41] and RTRs [42] has been hampered by a limited sample size. The adequately powered VITAL-study (n = 281) failed to show significant effects of paricalcitol on inflammatory biomarkers (CRP, fibrinogen, interleukin 6, TNF)in type 2 diabetes mellitus [25]. Taken together, the results of interventional studies on vitamin D agonist treatment seem inconclusive.

In the present study we found no evidence of paricalcitol influencing markers reflecting monocyte/macrophage (i.e. sCD163, sCD14, neopterin) or neutrophil (i.e. NGAL) activation, which signals no major clinically beneficial effect of paricalcitol on the activation and interplay of immune cells in the context of kidney transplantation. However, in light of recent findings suggesting that paricalcitol modulates inflammatory responses by influencing the calcineurin-axis [36], is it possible that anti-inflammatory effects could be masked by calcineurin inhibitor treatment, the corner stone in the immunosuppressive regimen for all our study participants.

Paricalcitol appears also to reduce development of renal interstitial fibrosis in obstructive nephropathy [43] and RTRs [44]. Metalloproteases, including MMP-9, are major regulators of ECM protein metabolism [45]. One might be tempted to interpret the trend towards reduced MMP-9 in paricalcitol-treated patients as a potential effect of VDRA on tissue extracellular matrix (ECM) remodeling, but it remains a speculation. A major source of MMP-9 in the circulation is neutrophils [46], hence plasma levels might also to some extent reflect neutrophil activation status.

We found that patients randomized to paricalcitol experienced an *increase* in circulating levels of OPG not seen in the control group. Correspondingly, the expression of the gene TNFRSF11B coding for OPG was also higher in renal graft tissue of patients in the intervention group. The result is consistent with experimental data on the immunomodulatory effects of 1,25-hydroxyvitamin D3 [47] and a similar clinical trial in hemodialysis patients [48]. Hansen *et al.* [48] found the rise in OPG in patients treated with paricalcitol to be correlated with the degree of suppression of PTH, partly explaining their results. Such a correlation was not clear in our cohort, despite a significant PTH-lowering effect of paricalcitol [31]. OPG protects the skeleton from excessive bone resorption by attaching to receptor activator of nuclear factor kappa-B ligand (RANKL) and preventing it from binding to its receptor on osteoclasts, RANK [49]. Plasma OPG has been suggested as a stable marker of the general activity in the RANKL/RANK system, a system that is linked to fibrogenesis and regulation of extracellular matrix. It is debated whether OPG itself is cardioprotective or a reactive proinflammatory molecule [50, 51]but modulatory roles in vascular injury and calcification, systemic inflammation and atherosclerosis, as well as in fibrosis pathways have been suggested [52, 53]. Thus, together with the potential downregulation of MMP-9 the effect seen on OPG may suggest that paricalcitol could have some effect on fibrogenesis in RTRs.

## Interpretation of findings

Since inflammatory markers typically have a wide distribution and relatively large SD's, the power to detect group differences in a moderate-sized study, such as ours, could be lower than

anticipated. As an example, MMP-9-levels were reduced by mean 25 ng/ml in the intervention group, while the controls had a 2 ng/ml rise during the study period. However, a 45 ng/ml difference in treatment effect would be needed to claim statistical significance (p <0.05). This is a high threshold for a biomarker whose reference range in healthy males is approximately 20–100 (M ±2SD) ng/ml [54]. Effect sizes may be more relevant that any p-value in itself [55]. Thus, although significant p-values are lacking in the current study, our results should not be interpreted as firm evidence against a potential anti-inflammatory effect of paricalcitol. Instead our results signal possible small-to-moderate effects within the limits of the reported confidence intervals. Taken together, evidence nevertheless seems too inconsistent to motivate the routine use of VDRA to reduce inflammation or improve vascular health in the transplant population. However, the interaction between paricalcitol and OPG as seen both at protein and transcript levels should be further explored as a potential important target for VDRAs. Also, though considered beyond the scope of the current investigation, the potential effect of paricalcitol on markers of oxidative stress (e.g. lipid peroxidation metabolites), as well as inflammatory metabolites (e.g. colonic microbiota-derived uraemic retention solutes) would be an interesting focus for future studies.

## Strengths and limitations

There was a high level of adherence to treatment in the paricalcitol group and no patient-initiated withdrawals [31]. The study population has been well characterized. Both circulating biomarker levels and tissue biomarker gene expression were evaluated, thus increasing the robustness of the results. However, sample size was calculated for the primary trial endpoint, not for the detection of potential treatment effects on levels of inflammatory markers. Notably, this study is of an explorative nature, testing many biomarkers at the same time. The suggested association between VDRA and OPG must be interpreted with caution, due to the possibility of making type 1-errors when performing multiple statistical tests. If strict Bonferroni correction for multiple testing was to be applied in this study, a p-value of <0,003 (0,05/15) would be needed to demonstrate statistical significance. Conclusions drawn from this study might only be valid for a white European population of RTRs with a reasonably good allograft function (i.e. eGFR>30 ml/min). Results are not necessarily applicable for recipients with vitamin D deficiency. We acknowledge that investigational bias might be a problem in open label trials, but for administrative reasons placebo drugs were unfortunately not available.

## Conclusions

In newly transplanted RTRs with adequate graft function, we were not able to demonstrate convincing reductions in levels of circulating biomarkers of inflammation and endothelial function after ten months of paricalcitol treatment. If present, a modulating effect of VDRA-treatment on systemic inflammation in this patient group is likely to be modest. We found that VDRA-treatment might increase levels of OPG, both in the circulation and in renal tissue, but this result needs to be replicated and validated.

## Supporting information

**S1 Table. Results for the per-protocol population.**
(DOCX)

**S2 Table. Results for patients with no rejection during the study period.** Results for patients with non-inflammatory cause of end-stage kidney disease.
(DOCX)

**S1 File. Project protocol.**
(DOC)

**S2 File. Consort checklist for main trial.**
(DOC)

**S3 File. Consort diagram.**
(BMP)

## Acknowledgments

Thanks to Dr. My Hanna Sofia Svensson for her great contribution to the design of the study. Thanks to study participants, study nurses and bioengineers at the Kidney Physiology Research Laboratory.

## Author Contributions

**Conceptualization:** Hege Kampen Pihlstrøm, Thor Ueland, Pål Aukrust, Geir Mjøen, Karsten Midtvedt, Anders Hartmann, Hallvard Holdaas.

**Data curation:** Hege Kampen Pihlstrøm, Thor Ueland, Annika E. Michelsen, Franscesca Gatti, Clara Hammarström, Monika Kasprzycka, Junbai Wang, Ivar Anders Eide.

**Formal analysis:** Hege Kampen Pihlstrøm, Franscesca Gatti, Clara Hammarström, Monika Kasprzycka, Junbai Wang.

**Funding acquisition:** Hege Kampen Pihlstrøm, Anders Hartmann, Hallvard Holdaas.

**Investigation:** Hege Kampen Pihlstrøm, Thor Ueland, Franscesca Gatti, Clara Hammarström, Monika Kasprzycka, Junbai Wang, Geir Mjøen, Dag Olav Dahle, Ivar Anders Eide, Hallvard Holdaas.

**Methodology:** Hege Kampen Pihlstrøm, Thor Ueland, Annika E. Michelsen, Pål Aukrust, Franscesca Gatti, Clara Hammarström, Monika Kasprzycka, Junbai Wang, Guttorm Haraldsen, Geir Mjøen, Dag Olav Dahle, Ivar Anders Eide, Anders Hartmann, Hallvard Holdaas.

**Project administration:** Hege Kampen Pihlstrøm, Monika Kasprzycka, Guttorm Haraldsen, Karsten Midtvedt, Ivar Anders Eide, Anders Hartmann, Hallvard Holdaas.

**Resources:** Thor Ueland, Pål Aukrust, Franscesca Gatti, Monika Kasprzycka, Junbai Wang.

**Software:** Junbai Wang.

**Supervision:** Thor Ueland, Pål Aukrust, Guttorm Haraldsen, Karsten Midtvedt, Hallvard Holdaas.

**Visualization:** Junbai Wang.

**Writing – original draft:** Hege Kampen Pihlstrøm, Thor Ueland, Annika E. Michelsen, Pål Aukrust, Junbai Wang, Geir Mjøen, Karsten Midtvedt.

**Writing – review & editing:** Hege Kampen Pihlstrøm, Thor Ueland, Annika E. Michelsen, Pål Aukrust, Franscesca Gatti, Clara Hammarström, Monika Kasprzycka, Junbai Wang, Guttorm Haraldsen, Geir Mjøen, Dag Olav Dahle, Karsten Midtvedt, Ivar Anders Eide, Anders Hartmann, Hallvard Holdaas.

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
