## [Decision Letter · Decision Letter 0]

7 Sep 2020

PONE-D-20-18424

Exploring the potential effect of paricalcitol on markers of inflammation in de novo renal transplant recipients

PLOS ONE

Dear Dr. Pihlstrøm,

Thank you for submitting your manuscript to PLOS ONE. After careful consideration, we feel that it has merit but does not fully meet PLOS ONE’s publication criteria as it currently stands. Therefore, we invite you to submit a revised version of the manuscript that addresses the points raised during the review process.

ACADEMIC EDITOR:

Generally well-executed study albeit with quite a few limitations (acknowledged by authors), but seems to have merit if reviewers' concerns can be adequately addressed by authors.

We look forward to receiving your revised manuscript.

Kind regards,

Frank JMF Dor, M.D., Ph.D., FEBS, FRCS

Academic Editor

PLOS ONE

Journal Requirements:

2. PLOS ONE requires that methods are described in enough detail to allow suitably skilled investigators to fully replicate and evaluate the study. Please provide more details on the methods and sample types used to quantify the inflammatory markers in your study (i.re, RNA quantification and ELISA experiments) and all sources and catalog numbers of reagents and ELISA kits used to measure them.

Please also clarify which tests were run at the one year visit. If a questionnaire was used during the visit, ensure that you have provided sufficient details that others could replicate the analyses. For instance, if you developed a questionnaire as part of this study and it is not under a copyright more restrictive than CC-BY, please include a copy, in both the original language and English, as Supporting Information.

Reviewers' comments:

Reviewer's Responses to Questions

**Comments to the Author**

1. Is the manuscript technically sound, and do the data support the conclusions?

Reviewer #1: Yes

Reviewer #2: Yes

2. Has the statistical analysis been performed appropriately and rigorously? 

Reviewer #1: Yes

Reviewer #2: No

3. Have the authors made all data underlying the findings in their manuscript fully available?

Reviewer #1: Yes

Reviewer #2: Yes

4. Is the manuscript presented in an intelligible fashion and written in standard English?

Reviewer #1: Yes

Reviewer #2: Yes

5. Review Comments to the Author

Reviewer #1: The authors examined markers of inflammation in patients with a recent renal transplant in a RCT involving the use of paricalcitol to see if paricalcitol would aid in the reduction of inflammation.

The study was designed to investigate a different primary outcome (which the authors have published separately). Here, they analysis the results of cytokine analysis as well as microarray analysis.

All statistical analyses seem appropriately performed. Table 1 & 2 are nicely put together, with easily identifiable information provided in the table description/legend. It is clear what data is normally distributed and what was not. It owuld be nice when the authors present a result in the text with a p-value to also include which type of statistical test was run to obtain this p-value. This will make the manuscript even more reproducible.

Figure 2 in the version that printed for this reviewer does not contain the gene name information. The powerpoint linked in the online version, does. It should be made certain that the gene information appears in the final version, as this is important information to convey to the reader.

The authors did a nice job of describing study limitations.

Reviewer #2: Generally well written and adequate interpretation of a study with many limitations as the authors have pointed out.

I note the following major points -

1. The authors are measuring inflammatory markers post transplantation and they could have looked at the effect of paricalcitol on inflammatory metabolites such as metabolites eg TMAO, p-cresyl sulphate (PCS), p-cresyl glucuronide (PCG), indoxyl sulphate (IS).

2. There is quite a bit of vague terminology

e.g. in the 1st paragraph - "inflammatory imbalance in the immune system". What does this actually mean?

"interleukin-6 (IL-6) and its major product, CRP" - CRP is not a product of Il-6, instead Il-6 can control hepatic CRP generation.

3. Is the statistical analysis for data in table 2 performed with ANCOVA? This would be more appropriate than multiple t-tets. How did the authors control for multiple statistical testing; could they have applied a Bonferroni correction?

4. Soluble CD25 is not a well validated marker for T cell activation. I would remove this data unless they can demonstrate CD25 expression on isolated T cells by FACS.

5. In discussion, "evidence nevertheless seems too inconsistent to motivate the routine use of VDRA to reduce inflammation or improve vascular health in the transplant population". How could the current study be adapted to answer this?

6. PLOS authors have the option to publish the peer review history of their article (what does this mean?). If published, this will include your full peer review and any attached files.

Reviewer #1: No

Reviewer #2: No

---

## [Author Response · Author response to Decision Letter 0]

28 Oct 2020

A: Journal Requirements:

ANSWER: We have gone through the requirements once more and hopefully corrected any examples of lacking/incorrect information.

2. PLOS ONE requires that methods are described in enough detail to allow suitably skilled investigators to fully replicate and evaluate the study. Please provide more details on the methods and sample types used to quantify the inflammatory markers in your study (i.re, RNA quantification and ELISA experiments) and all sources and catalog numbers of reagents and ELISA kits used to measure them.

ANSWER: Thank you for pointing out this requirement! We have now included a more elaborate description of the enzyme immunoassay in the methods section of the manuscript As for the gene expression analysis, an elaborate method description is to be found in the main study publication (ref 29)., which we have cited in the current manuscript. Do you still want it repeated explicitly in the text? 

Please also clarify which tests were run at the one year visit. If a questionnaire was used during the visit, ensure that you have provided sufficient details that others could replicate the analyses. For instance, if you developed a questionnaire as part of this study and it is not under a copyright more restrictive than CC-BY, please include a copy, in both the original language and English, as Supporting Information.

ANSWER: The original publication listed as ref 29 (Pihlstrom et al, Transplant International 2017, PMID: 28436117) describes in detail what tests were performed at the 1-year visit. On your request we have included a sentence briefly summing up these investigations in the “study design” chapter. For the current analyses, the only relevant “tests” are the biomarker analyses, which were performed en bloc at our Research Laboratory as described above. No questionnaire was part of the data collection process. 

3. We note that you have indicated that data from this study are available upon request. PLOS only allows data to be available upon request if there are legal or ethical restrictions on sharing data publicly. For information on unacceptable data access restrictions, please see http://journals.plos.org/plosone/s/data-availability#loc-unacceptable-data-access-restrictions. In your revised cover letter, please address the following prompts:

 ANSWER: Unfortunately there are ethical and legal restrictions preventing us from uploading data to public repositories or including the full dataset as Supplementary Material. Norwegian kidney transplant recipients receiving their transplant in the time-frame of the current study belong to a relatively small group, and very little personal data would be needed in order to indirectly identify individual study participants. We have been in dialogue with the Data Protection Authority of Oslo University Hospital in this matter, and here is their response: 

"CONCERNING SHARING OF RESEARCH DATA

Pursuant to Regulation (EU) No. 2016/679, General Data Protection Regulation (GDPR) article 37, the designated Data Protection Officer at Oslo University Hospital (OUS) is appointed. The controller and the processor shall ensure that the data protection officer is involved, properly and in a timely manner, in all issues which relate to the protection of personal data, cf. GDPR article 38. 

According to GDPR article 9, the processing of genetic data or health data shall be prohibited unless the data subject has given an explicit consent to the processing of those personal data for one or more specified purposes. Personal data is defined as any information relating to an identified or identifiable natural person; an identifiable natural person is one who can be identified, directly or indirectly. 

Consequently, depositing de-identified data in a public, community-supported repository in order to submit an article is not considered compliant with EU and Norwegian law in this matter. In order to comply with the relevant legislations, data would need to be fully anonymized.

If required, provisions can be made for the inspection of the data as long as the data is under the hospital's control, hence the Data Controller-responsibility".

The implication of the above described regulations is that any reader interested in inspecting the data may prompt this request to the OUS Data Protection Officer, Tor Åsmund Martinsen (personvern@oslo-universitetssykehus.no). The corresponding author, Hege Kampen Pihlstrøm (hegphi@ous-hf.no) should also be contacted. A de-identified study data file may then be made available. 

ANSWER: We appreciate the policy that the findings referred to in the manuscript should be available in more detail than what we have provided. Hence we have chosen to include supplementary tables (1-2) for sensitivity analyses and added them to the submission as Supporting Information Files. 

Reviewer #1: The authors examined markers of inflammation in patients with a recent renal transplant in a RCT involving the use of paricalcitol to see if paricalcitol would aid in the reduction of inflammation.

The study was designed to investigate a different primary outcome (which the authors have published separately). Here, they analysis the results of cytokine analysis as well as microarray analysis.

All statistical analyses seem appropriately performed. Table 1 & 2 are nicely put together, with easily identifiable information provided in the table description/legend. It is clear what data is normally distributed and what was not. It would be nice when the authors present a result in the text with a p-value to also include which type of statistical test was run to obtain this p-value. This will make the manuscript even more reproducible.

ANSWER: Reference to the tests used has been added in the text!

Figure 2 in the version that printed for this reviewer does not contain the gene name information. The powerpoint linked in the online version, does. It should be made certain that the gene information appears in the final version, as this is important information to convey to the reader. 

ANSWER: We will ensure the final figure 2 contains the gene information! Thank you for pointing out this lapse in the version you received. The figure has also been revised (removal of genes related to CD25 as recommended by reviewer 2), and we have included an annotation in the figure legend for genes in the heat map which are known by different names than the proteins they encode.

The authors did a nice job of describing study limitations.

Reviewer #2: Generally well written and adequate interpretation of a study with many limitations as the authors have pointed out.

I note the following major points -

1. The authors are measuring inflammatory markers post transplantation and they could have looked at the effect of paricalcitol on inflammatory metabolites such as metabolites eg TMAO, p-cresyl sulphate (PCS), p-cresyl glucuronide (PCG), indoxyl sulphate (IS).

ANSWER: Thank you for the valuable comment! It would indeed have been interesting to investigate inflammatory metabolites as well as the selected circulating biomarkers. However, we are already close to the limit of parameters possible to investigate in this dataset due to its limited size. Hence we feel the analyses of colonic microbiota-derived uraemic retention solutes would be a great idea for a future study, but unfortunately out of the scope of the current project. A comment has been included in the discussion section, suggesting these inflammatory metabolites as focus for upcoming studies.

2. There is quite a bit of vague terminology

e.g. in the 1st paragraph - "inflammatory imbalance in the immune system". What does this actually mean?

"interleukin-6 (IL-6) and its major product, CRP" - CRP is not a product of Il-6, instead Il-6 can control hepatic CRP generation.

ANSWER: Thank you for pointing out imprecise terminology in the manuscript. We have rephrased some sentences in the introduction section according to your comments. Hopefully they will serve as clarification.

3. Is the statistical analysis for data in table 2 performed with ANCOVA? This would be more appropriate than multiple t-tets. How did the authors control for multiple statistical testing; could they have applied a Bonferroni correction?

ANSWER: Table 2 presented results of t-tests. As described in the methods section, ANCOVA was performed as a sensitivity analysis. Editors have made it clear that all data which is part of the study should be presented/available for the reader, hence we have included the ANCOVA analyses in a separate column in table 2. Results are comparable to the t-test results.

With reference to the two- tailed P-value of ≤0.05 commonly used as cut-off for statistical significance, a p-value of ≤0,003 (0,05/15) would be needed to demonstrate statistical significance after Bonferroni correction for multiple testing in this study. We believe that we have already made it a focus point in the discussion that the study lacks power to conclude firmly on any anti-inflammatory treatment effects, and that the suggested influence of paricalcitol treatment on OPG should be explored in larger studies. We have now added a sentence acknowledging the issue of multiple testing in the limitations section. 

4. Soluble CD25 is not a well validated marker for T cell activation. I would remove this data unless they can demonstrate CD25 expression on isolated T cells by FACS.

ANSWER: We have removed the data on CD25 as you suggested.

5. In discussion, "evidence nevertheless seems too inconsistent to motivate the routine use of VDRA to reduce inflammation or improve vascular health in the transplant population". How could the current study be adapted to answer this?

ANSWER: To be adequately powered to help us conclude on the potential treatment effect of paricalcitol on inflammation in renal transplant recipients, the study should have been larger, with a longer follow-up time. Being a post-hoc analysis of a study designed to investigate a different primary endpoint, we do not have the option of extending the duration of the study or include more patients. We have tried to underscore these limitations at the end of the discussion section. 

FINAL REMARKS: One of the coauthors, My H S Svensson, did not feel that she had contributed enough to the study and the manuscript to deserve authorship, and consequently we have removed her from the list of authors.

---

## [Decision Letter · Decision Letter 1]

30 Nov 2020

Exploring the potential effect of paricalcitol on markers of inflammation in de novo renal transplant recipients

PONE-D-20-18424R1

Dear Dr. Pihlstrøm,

We’re pleased to inform you that your manuscript has been judged scientifically suitable for publication and will be formally accepted for publication once it meets all outstanding technical requirements.

Kind regards,

Frank JMF Dor, M.D., Ph.D., FEBS, FRCS

Academic Editor

PLOS ONE

Additional Editor Comments (optional):

Reviewers' comments:

Reviewer's Responses to Questions

**Comments to the Author**

1. If the authors have adequately addressed your comments raised in a previous round of review and you feel that this manuscript is now acceptable for publication, you may indicate that here to bypass the “Comments to the Author” section, enter your conflict of interest statement in the “Confidential to Editor” section, and submit your "Accept" recommendation.

Reviewer #1: All comments have been addressed

Reviewer #2: All comments have been addressed

2. Is the manuscript technically sound, and do the data support the conclusions?

Reviewer #1: (No Response)

Reviewer #2: Yes

3. Has the statistical analysis been performed appropriately and rigorously? 

Reviewer #1: (No Response)

Reviewer #2: Yes

4. Have the authors made all data underlying the findings in their manuscript fully available?

Reviewer #1: (No Response)

Reviewer #2: Yes

5. Is the manuscript presented in an intelligible fashion and written in standard English?

Reviewer #1: (No Response)

Reviewer #2: Yes

6. Review Comments to the Author

Reviewer #1: (No Response)

Reviewer #2: All my comments have been addressed and no further comments to add from my side. I accept the revised submission

7. PLOS authors have the option to publish the peer review history of their article (what does this mean?). If published, this will include your full peer review and any attached files.

Reviewer #1: No

Reviewer #2: No

---

## [Editor Report · Acceptance letter]

3 Dec 2020

PONE-D-20-18424R1 

Exploring the potential effect of paricalcitol on markers of inflammation in *de novo* renal transplant recipients 

Dear Dr. Pihlstrøm:

I'm pleased to inform you that your manuscript has been deemed suitable for publication in PLOS ONE. Congratulations! Your manuscript is now with our production department. 

Kind regards, 

on behalf of

Dr. Frank JMF Dor 

Academic Editor

PLOS ONE